# Magnetic properties of a multicomponent intermetallic compound $Tb_{0.25}Dy_{0.25}Ho_{0.25}Er_{0.25}Al_2$

P. K. Jesla[1], Jeyaramane Arout Chelvane[2] and R. Nirmala[1*]

**1** Department of Physics, Indian Institute of Technology Madras, Chennai 600 036 India
**2** Defence Metallurgical Research Laboratory, Hyderabad 500 058, India

* nirmala@physics.iitm.ac.in

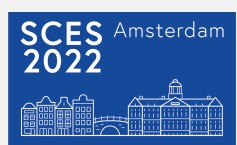 *International Conference on Strongly Correlated Electron Systems (SCES 2022)*
## Abstract

Polycrystalline multicomponent Laves phase rare earth intermetallic compound $Tb_{0.25}Dy_{0.25}Ho_{0.25}Er_{0.25}Al_2$ has been synthesized by arc-melting and characterized by powder X-ray diffraction, magnetization and heat capacity measurements. The sample has $MgCu_2$-type cubic structure (space group Fd-3m) at 300 K. The compound $Tb_{0.25}Dy_{0.25}Ho_{0.25}Er_{0.25}Al_2$ orders ferromagnetically at 50 K ($T_C$). This value is almost equal to the average of the ferromagnetic ordering temperatures of the individual $RAl_2$ (R = Tb, Dy, Ho and Er) compounds. Field dependence of magnetization below $T_C$ indicates soft-ferromagnetic nature. From the magnetization vs field data obtained near $T_C$ and the temperature dependence of heat capacity, low field magnetocaloric effect has been estimated.

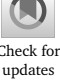
## 1 Introduction

Rare-earth intermetallic compounds $RAl_2$ (R= Gd, Tb, Dy, Ho and Er) crystallize in $MgCu_2$-type, cubic Laves phase structure at room temperature and display a second-order paramagnetic to ferromagnetic magnetic transition at Curie temperature ranging from 13 K (for $ErAl_2$) to 167 K (for $GdAl_2$) [1]. These $RAl_2$ compounds are well-known as model systems to study the influence of magnetic exchange interaction, crystalline electric field and magnetic anisotropy on the magnetocaloric effect (MCE). Several binary and pseudo-binary $RAl_2$ compounds display large MCE around the magnetic transition temperature [2] [3]. Using three or more magnetic rare earth elements in the rare-earth sublattice will help to modify Curie temperature and also magnetocrystalline anisotropy [4]. Large MCE spread over a wide temperature range may also result. Reports on multicomponent systems and high entropy alloys [5] [6] have motivated us to study the magnetic properties of a multicomponent $RAl_2$ system where R site is occupied by four equimolar heavy rare earth elements.

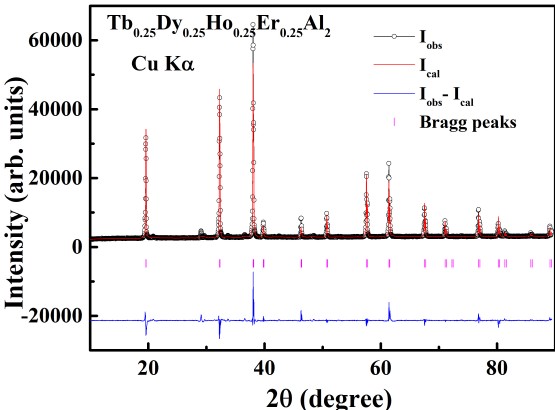

Figure 1: Powder X-ray diffraction data of $Tb_{0.25}Dy_{0.25}Ho_{0.25}Er_{0.25}Al_2$ and its Rietveld refinement

## 2 Experimental details

Polycrystalline $Tb_{0.25}Dy_{0.25}Ho_{0.25}Er_{0.25}Al_2$ compound has been prepared by arc melting under argon atmosphere, starting from stoichiometric amounts of pure elements. The sample was remelted several times to improve homogeneity. The sample was characterized using powder X-ray diffraction (XRD) at 300 K (Cu-K$\alpha$) followed by energy dispersive X-ray analysis (EDX). Magnetization (M) measurements have been performed using a vibrating sample magnetometer (Lake Shore 7410-S) in fields up to 15 kOe in a temperature (T) range of 15 K to 300 K. Heat capacity (C) has been measured as a function of temperature from 2 K to 300 K by relaxation technique using a physical property measurement system (PPMS, Quantum Design) [7].

## 3 Results and discussion

Powder X-ray diffraction data confirm that the sample crystallizes in an $MgCu_2$-type structure (cubic, space group Fd-3m) [Figure 1]. The lattice parameter ($a$) obtained by Rietveld refinement of the powder XRD data is 7.8394(3) Å. The value is close to the mean value of the lattice parameters of $TbAl_2$, $DyAl_2$, $HoAl_2$ and $ErAl_2$ [Table 1]. The compound $Tb_{0.25}Dy_{0.25}Ho_{0.25}Er_{0.25}$ $Al_2$ is found to be isostructural with the parent compounds $RAl_2$. Nominal composition of the sample is verified by the EDX analysis.

Magnetization measurements of $Tb_{0.25}Dy_{0.25}Ho_{0.25}Er_{0.25}Al_2$ have been carried out in zero-field-cooled (ZFC) and field-cooled (FC) modes during warming from 15 K to 300 K in a field of 5 kOe. The M(T) data show a transition from paramagnetic to ferromagnetic state at 50 K ($T_C$) [Figure 2(a)]. The ferromagnetic ordering temperature has been identified by plotting first derivative of magnetization with respect to temperature [Inset in Figure 2(a)]. No thermomagnetic irreversibility between ZFC and FC magnetization is observed. The compounds $TbAl_2$, $DyAl_2$, $HoAl_2$ and $ErAl_2$ order ferromagnetically at about 105 K, 65 K, 29 K and 13 K respectively [8]. Interestingly, the mean value of these ordering temperatures is nearly equal to the ordering temperature of the multicomponent system under study. Similar behaviour has also been observed in five-component Laves phase intermetallic compound $Gd_{0.2}Tb_{0.2}Dy_{0.2}Ho_{0.2}Er_{0.2}Al_2$ [9].

Paramagnetic susceptibility obeys Curie-Weiss law. From the fit, effective paramagnetic moment ($\mu_{eff}$) and the paramagnetic Curie temperature ($\theta_p$) values are calculated as 10.1

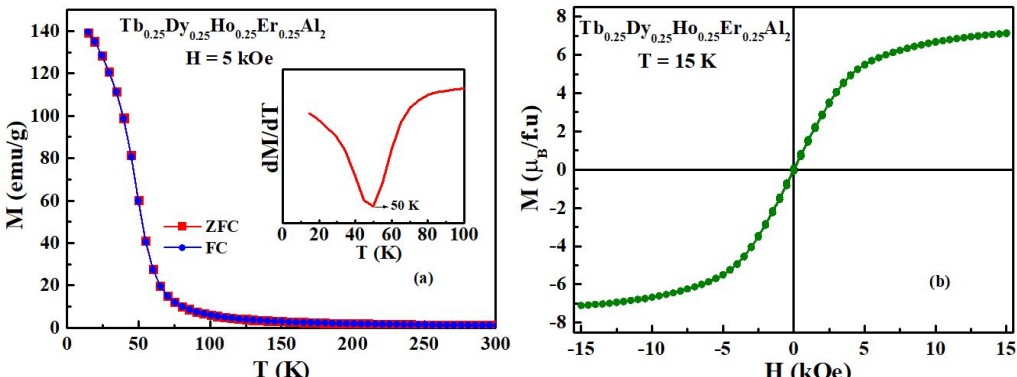

Figure 2: (a) Zero-field-cooled (ZFC) and field-cooled (FC) magnetization data of $Tb_{0.25}Dy_{0.25}Ho_{0.25}Er_{0.25}Al_2$ compound [Inset shows dM/dT vs T plot] (b) Magnetization vs field at 15 K

$\mu_B$ and 52 K respectively. The calculated $\mu_{eff}$ value is almost equal to average paramagnetic moment value of tripositive rare earth ions present (i.e. 10.2 $\mu_B$ ). The dominance of ferromagnetic interactions is confirmed from the positive $\theta_p$ value. Magnetization vs field (M-H) data obtained at 15 K in fields up to 15 kOe show soft-ferromagnetic behavior. The saturation magnetization value is obtained by extending magnetization to infinite field (i.e. M in the limit $1/H \to 0$) is 9.0 $\mu_B$. This value is less than that expected for collinear alignment of rare earth moments in the compound (9.5 $\mu_B$). Isothermal magnetization of $Tb_{0.25}Dy_{0.25}Ho_{0.25}Er_{0.25}Al_2$ has been measured as a function of magnetic field from 15 K to 80 K with a temperature interval of 5 K in fields up to 15 kOe [Figure 3(a)]. Magnetocaloric effect around $T_C$ has been calculated in terms of isothermal magnetic entropy change ($\Delta S_m$) from the M-H data using the equation

$$-\Delta S_m = \mu_0 \int_{H_i}^{H_f} \left( \frac{\partial M(T,H)}{\partial T} \right)_H dH, \tag{1}$$

obtained from the thermodynamic Maxwell relation [10]. Here, $H_i$ and $H_f$ stand for the initial and final applied field values and $\mu_0$ is the permeability of free space. The multicomponent system $Tb_{0.25}Dy_{0.25}Ho_{0.25}Er_{0.25}Al_2$ exhibits normal magnetocaloric effect below $T_C$ as expected in ferromagnetically ordered materials [Figure 3(b)]. The maximum isothermal magnetic entropy change ($\Delta S_m^{max}$) is about -3.6 $Jkg^{-1}K^{-1}$ at 47.5 K for a field change of 10 kOe. It is comparable with single component $RAl_2$ [Table 1]. The corresponding relative cooling power is large and is about 69 J/kg for 10 kOe field change. The second order nature of the transition has been confirmed by Arrott plots [Figure 3(c)].

Heat capacity of $Tb_{0.25}Dy_{0.25}Ho_{0.25}Er_{0.25}Al_2$ compound has been measured as a function of temperature in zero magnetic field and in applied field of 10 kOe. The data reveal a $\lambda$-type transition around $T_C$ in zero field [Figure 4(a)]. This corresponds to the ferromagnetic ordering observed in the magnetic data. However, the transition is not as sharp as the one observed in end member $RAl_2$ (R = Tb, Dy, Ho and Er) compounds. The peak gets diminished upon the application of 10 kOe field and shifts to higher temperature as evidenced in ferromagnets. At room temperature, heat capacity approaches the Dulong and Petit's limit (3NR). Isothermal magnetic entropy change is calculated from the heat capacity data using the equation

$$\Delta S_m = \int_0^T \frac{C(T,H) - C(T,0)}{T} dT, \tag{2}$$

and is compared with the value calculated using magnetization data [Figure 4(b)]. Magnetocaloric effect around $T_C$ has also been quantified in terms of adiabatic temperature change

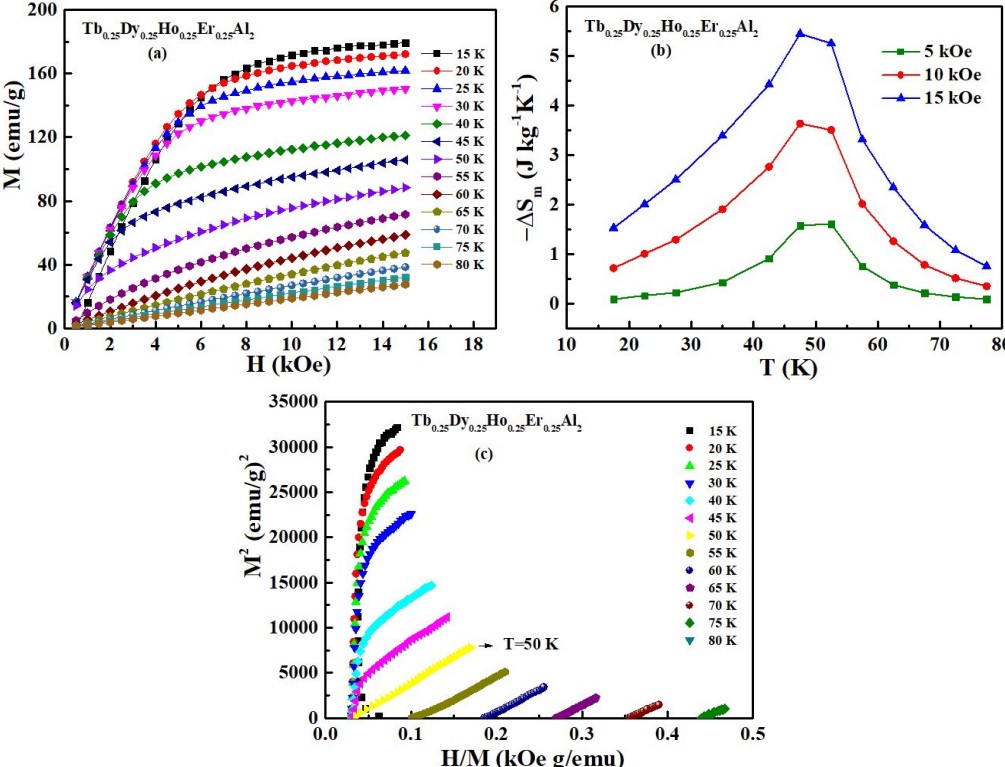

Figure 3: (a) Magnetization vs field isotherms in the temperature range of 15 K to 80 K, (b) isothermal magnetic entropy change vs temperature and (c) Arrott plots of $Tb_{0.25}Dy_{0.25}Ho_{0.25}Er_{0.25}Al_2$ compound.

($\Delta T_{ad}$). Zero field heat capacity data have been used along with temperature dependent magnetic entropy change in order to obtain adiabatic temperature change [Figure 4(c)]. Following equation has been used for the same [11]

$$\Delta T_{ad} = \frac{T}{C(T,0)} \Delta S_m .$$ (3)

The maximum adiabatic temperature change of about 2.4 K at T = 52.5 K has been observed for $Tb_{0.25}Dy_{0.25}Ho_{0.25}Er_{0.25}Al_2$ compound for 15 kOe magnetic field change. This is of the same order of adiabatic temperature change observed in the $RAl_2$ (R = Gd, Tb, Dy, Ho and Er) compounds. Thus, substitution of multiple principal elements at the rare earth site of $RAl_2$ compound helps in tuning the ferromagnetic transition temperature in accordance with the de Gennes factor [13]. Multicomponent $Tb_{0.25}Dy_{0.25}Ho_{0.25}Er_{0.25}Al_2$ forms in the same crystal structure and shows an average $T_C$ of individual rare earth containing $RAl_2$ compounds and substantial magnetocaloric effect.

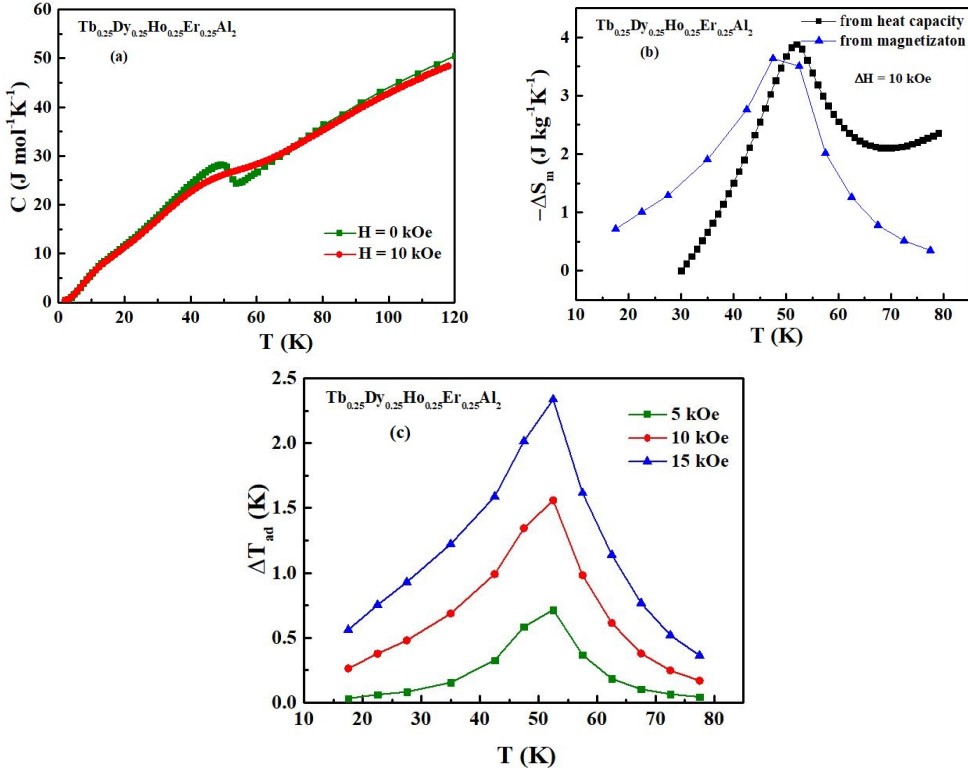

Figure 4: (a) Heat capacity vs temperature of $Tb_{0.25}Dy_{0.25}Ho_{0.25}Er_{0.25}Al_2$ compound in zero magnetic field and 10 kOe field, (b) isothermal magnetic entropy change vs temperature calculated using heat capacity and magnetization data and (c) adiabatic temperature change vs temperature for selected magnetic field changes.

Table 1: The comparison of structural, magnetic and magnetocaloric properties of $Tb_{0.25}Dy_{0.25}Ho_{0.25}Er_{0.25}Al_2$ with $RAl_2$ (R = Tb, Dy, Ho and Er) compounds.

| Compound | $a$ (Å) | $T_C$ ( K ) | $\Delta S_m^{max}$ ($Jkg^{-1}K^{-1}$) | $\Delta T_{ad}^{max}$ (K) | $\Delta H$ (kOe) | Reference |
|---|---|---|---|---|---|---|
| $TbAl_2$ | 7.867 | 105 | 5.5 | 2.2 | 20 | [8], [12] |
| $DyAl_2$ | 7.8370 | 65 | 9.7 | 3.6 | 20 | [8], [12] |
| $HoAl_2$ | 7.8182 | 29 | 12.8 | 4.6 | 20 | [8], [12] |
| $ErAl_2$ | 7.7957 | 13 | 22.6 | | 20 | [8], [12] |
| $Tb_{0.25}Dy_{0.25}Ho_{0.25}Er_{0.25}Al_2$ | 7.8394(3) | 50 | 5.5 | 2.4 | 15 | This work |

# 4  Conclusions

Multicomponent rare earth intermetallic compound $Tb_{0.25}Dy_{0.25}Ho_{0.25}Er_{0.25}Al_2$ crystallizes in cubic structure and orders ferromagnetically around 50 K. Magnetocaloric effect has been calculated using both the heat capacity and magnetization data and the compound shows a moderate magnetocaloric effect. A large isothermal entropy change of about -3.6 $J/Kg^{-1}K^{-1}$ at 47.5 K is observed for a field change of 10 kOe.

# 5 Acknowledgements

P. K. J thanks DST-INSPIRE for fellowship. Physical property measurement system used in this work was supported by DST, India under FIST program (No. SR/FST/PSII-038/2016).

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
