# Peer review of "Magnetic properties of a multicomponent intermetallic compound Tb0.25Dy0.25Ho0.25Er0.25Al2"

_SciPost Physics Proceedings, doi:SciPost Phys. Proc. 11, 013 (2023)_

## Round 1 · Referee Report · Anonymous (Referee 1) · 2023-1-7

Strengths

1: Complete analysis of multi-component rare-earth inter metallic with an eye towards magneto-caloric effect.

Weaknesses

Presents data but does not go much beyond the basics.

Report

This paper meets the criteria for the SCES conference proceedings.

Requested changes

This is an overall well written and complete study of a multi component inter-metallic. I have a few small comments that will improve reading for non-experts.

1: Section 2: The acronym EDAX is introduced. I don’t see where the A is coming from. I believe it is generally abbreviated to EDX.
2: Is there a reference for the relaxation technique used to measure the specific heat.
3: On page 2 it is written that the saturation magnetization is 9.5 Bohr magnetron. Looking at Fig. 2b, it seems to be 7.5 mu_B/f.u. Please explain.
4: Similarly, it is stated that this is less than expected for collinear alignment. Please add the expected value.
5: The MCE of this compound is called ‘normal’, ‘substantial’ and ‘moderate’. These are rather qualitative and seem to be inconsistent with each other. It would help the reader to know what defines this and how it compares to other materials.
6: On page 4, just above Eq. 2 it is written ’…and shifts to the right as evidenced in ferromagnetic.’ Please change to ‘…shifts to higher temperature…’
7: A reference or explanation of Eq. 3 would be appreciated.
8: It would be useful to have comparison values of the adiabatic temperature changes for RAl2 compounds. Either a reference or maybe add them to table 1?
9: The sentence ‘Thus, substitution of multi-component…’ is unclear. What is the de Gennes factor and how does it connect to having multiple principal elements?

  • validity: high
  • significance: good
  • originality: ok
  • clarity: good
  • formatting: reasonable
  • grammar: good

Author:  Jesla P K  on 2023-01-25  [id 3271]

(in reply to Report 1 on 2023-01-07)
Category:
answer to question

Answers for the reviewer’s questions:
1. Section 2: The acronym EDAX is introduced. I don’t see where the A is coming from. I believe it is generally abbreviated to EDX.
Energy dispersive analysis of X-rays is abbreviated either as EDX or EDAX in literature. We now use EDX in the manuscript.

2. Is there a reference for the relaxation technique used to measure the specific heat?
Yes, we have now added the following reference. J.C. Lashley, M.F. Hundley, A. Migliori, J.L. Sarrao, P.G. Pagliuso, T.W. Darling, M. Jaime, J.C. Cooley, W.L. Hults, L. Morales, D.J. Thoma, J.L. Smith, J. Boerio-Goates, B.F. Woodfield, G.R. Stewart, R.A. Fisher, N.E. Phillips, Critical examination of heat capacity measurements made on a Quantum Design physical property measurement system, Cryogenics 43(6), 369(2003) doi:10.1016/S0011-2275(03)00092-4.

3. On page 2 it is written that the saturation magnetization is 9.5 Bohr magnetron. Looking at Fig. 2b, it seems to be 7.5 mu_B/f.u. Please explain.

Saturation magnetization is calculated from M vs H data by extending magnetization to infinite magnetic field. It is the value of M when 1/H tends to zero. The value obtained is 9 Bohr magneton. This is now correctly mentioned in the manuscript.

4. Similarly, it is stated that this is less than expected for collinear alignment. Please add the expected value.
The value expected for collinear alignment is 9.5 Bohr magneton.

5. The MCE of this compound is called ‘normal’, ‘substantial’ and ‘moderate’. These are rather qualitative and seem to be inconsistent with each other. It would help the reader to know what defines this and how it compares to other materials.
Magnetocaloric effect is called ‘normal’ if -ΔSM is positive (as observed in ferromagnets) and it is called ‘inverse magnetocaloric effect’ if -ΔSM is negative (as observed in antiferromagnets). The -ΔSM value for 15 kOe field change is comparable with that of the individual RAl2 (R = Tb, Dy, Ho and Er) and these are moderate when compared to that of giant magnetocaloric materials.

6. On page 4, just above Eq. 2 it is written ’…and shifts to the right as evidenced in ferromagnetic.’ Please change to ‘…shifts to higher temperature…’

Yes, this change has been done.

7. A reference or explanation of Eq. 3 would be appreciated.
A reference [M. Foldeaki, et al., J. Appl. Phys. 82 (1) (1997) 309] is now added.
8. It would be useful to have comparison values of the adiabatic temperature changes for RAl2 compounds. Either a reference or maybe add them to table 1?
Yes this is added to table 1 in the revised manuscript.

Compound a
(Å) TC (K) ΔSmax
(Jkg-1K-1) ΔTad (K) ΔH (kOe) Reference
TbAl2 7.867 105 5.5 2.2 20 [8], [12]
DyAl2 7.8370 65 9.7 3.6 20 [8], [12]
HoAl2 7.8182 29 12.8 4.6 20 [8], [12]
ErAl2 7.7957 13 22.6 20 [8], [12]
Tb0.25Dy0.25Ho0.25Er0.25Al2 7.8394 50 5.5 2.4 15 This work

9. The sentence ‘Thus, substitution of multi-component…’ is unclear. What is the de Gennes factor and how does it connect to having multiple principal elements?

Transition temperature of isostructural RAl2 compounds scales with de Gennes factor
G= (g-1)2 J(J+1) where g is the gyromagnetic ratio. The multi-component system also follows the de Gennes rule. We have now added this reference in the revised manuscript. R. M. Bozorth, Magnetic Properties of Compounds and Solid Solutions of Rare‐Earth Metals, Journal of Applied Physics 38, 1366 (1967); https://doi.org/10.1063/1.1709625.

Attachment:

Answers_for_the_questions_22Jan2023_1.pdf

---

## Round 2 · Author Response

We thank the reviewer for the suggestions and here are the answers for the reviewer’s questions:
1. Section 2: The acronym EDAX is introduced. I don’t see where the A is coming from. I believe it is generally abbreviated to EDX.
Energy dispersive analysis of X-rays is abbreviated either as EDX or EDAX in literature. We now use EDX in the manuscript.

2. Is there a reference for the relaxation technique used to measure the specific heat?
Yes, we have now added the following reference. J.C. Lashley, M.F. Hundley, A. Migliori, J.L. Sarrao, P.G. Pagliuso, T.W. Darling, M. Jaime, J.C. Cooley, W.L. Hults, L. Morales, D.J. Thoma, J.L. Smith, J. Boerio-Goates, B.F. Woodfield, G.R. Stewart, R.A. Fisher, N.E. Phillips, Critical examination of heat capacity measurements made on a Quantum Design physical property measurement system, Cryogenics 43(6), 369(2003) doi:10.1016/S0011-2275(03)00092-4.

3. On page 2 it is written that the saturation magnetization is 9.5 Bohr magnetron. Looking at Fig. 2b, it seems to be 7.5 mu_B/f.u. Please explain.

Saturation magnetization is calculated from M vs H data by extending magnetization to infinite magnetic field. It is the value of M when 1/H tends to zero. The value obtained is 9 Bohr magneton. This is now correctly mentioned in the manuscript.

4. Similarly, it is stated that this is less than expected for collinear alignment. Please add the expected value.
The value expected for collinear alignment is 9.5 Bohr magneton.

5. The MCE of this compound is called ‘normal’, ‘substantial’ and ‘moderate’. These are rather qualitative and seem to be inconsistent with each other. It would help the reader to know what defines this and how it compares to other materials.
Magnetocaloric effect is called ‘normal’ if -ΔSM is positive (as observed in ferromagnets) and it is called ‘inverse magnetocaloric effect’ if -ΔSM is negative (as observed in antiferromagnets). The -ΔSM value for 15 kOe field change is comparable with that of the individual RAl2 (R = Tb, Dy, Ho and Er) and these are moderate when compared to that of giant magnetocaloric materials.

6. On page 4, just above Eq. 2 it is written ’…and shifts to the right as evidenced in ferromagnetic.’ Please change to ‘…shifts to higher temperature…’

Yes, this change has been done.

7. A reference or explanation of Eq. 3 would be appreciated.
A reference [M. Foldeaki, et al., J. Appl. Phys. 82 (1) (1997) 309] is now added.
8. It would be useful to have comparison values of the adiabatic temperature changes for RAl2 compounds. Either a reference or maybe add them to table 1?
Yes this is added to table 1 in the revised manuscript.

Compound a
(Å) TC (K) ΔSmax
(Jkg-1K-1) ΔTad (K) ΔH (kOe) Reference
TbAl2 7.867 105 5.5 2.2 20 [8], [12]
DyAl2 7.8370 65 9.7 3.6 20 [8], [12]
HoAl2 7.8182 29 12.8 4.6 20 [8], [12]
ErAl2 7.7957 13 22.6 20 [8], [12]
Tb0.25Dy0.25Ho0.25Er0.25Al2 7.8394 50 5.5 2.4 15 This work

9. The sentence ‘Thus, substitution of multi-component…’ is unclear. What is the de Gennes factor and how does it connect to having multiple principal elements?

Transition temperature of isostructural RAl2 compounds scales with de Gennes factor
G= (g-1)2 J(J+1) where g is the gyromagnetic ratio. The multi-component system also follows the de Gennes rule. We have now added this reference in the revised manuscript. R. M. Bozorth, Magnetic Properties of Compounds and Solid Solutions of Rare‐Earth Metals, Journal of Applied Physics 38, 1366 (1967); https://doi.org/10.1063/1.1709625.

---

## Editorial Decision

published